# Direct Inference of Cell Positions using Lens-Free Microscopy and Deep Learning

**Philipp Grüning**[*][1]                              GRUENING@INB.UNI-LUEBECK.DE
[1] *Institute for Neuro- and Bioinformatics, University of Lübeck, Germany*

**Falk Nette**[†][2]                              FALK.HENDRIK.NETTE@EMB.FRAUNHOFER.DE
[2] *Fraunhofer Research and Development Center for Marine and Cellular Biotechnology, Lübeck, Germany*

**Noah Heldt**[1]
**Ana Cristina Guerra de Souza**[2]
**Erhardt Barth**[1]                              BARTH@INB.UNI-LUEBECK.DE

## Abstract

With in-line holography, it is possible to record biological cells over time in a three-dimensional hydrogel without the need for staining, providing the capability of observing cell behavior in a minimally invasive manner. However, this setup currently requires computationally intensive image-reconstruction algorithms to determine the required cell statistics. In this work, we directly extract cell positions from the holographic data by using deep neural networks and thus avoid several reconstruction steps. We show that our method is capable of substantially decreasing the time needed to extract information from the raw data without loss in quality.

**Keywords:** semantic segmentation, deep learning, in-line holography

## 1. Introduction

Time-lapse microscopy (Kim, 2010) has become a pervasive tool for live-cell imaging since digital image sensors became widely available in the early years of this century. Compared to a conventional optical microscope, holographic imaging offers a number of advantages, including i.) the ability to build a lens-less imaging setup which reduces manufacturing costs substantially, ii.) a larger field of view, iii.) instantaneously capturing a three-dimensional object distribution in a single hologram, and iv.) the possibility to retrieve both amplitude and quantitative phase-shift profiles, the latter being particularly useful for observing usually near-transparent biological cells.

However, obtaining a utilizable image requires a computationally intensive reconstruction step that is based on the knowledge of light-matter interaction and wave interference. Accordingly, statistical analysis of a high-throughput cell-tracking experiment can take hours, which reduces the practicality of digital holography in this context.

Rivenson et al. (Rivenson et al., 2019) already showed that CNNs could learn holographic reconstruction. However, in our framework, no exact reconstruction is needed; we instead focus on a binary segmentation from which cell positions needed for cell-tracking can be derived.

---

[*] Contributed equally

[†] Contributed equally

## 1.1. Related Work

Tracking is an essential tool in the analysis of cell behavior, which in turn provides crucial information in a plethora of applications in medicine and biology. In general, object tracking (Li et al., 2013) can be divided into two different approaches: model evolution and tracking by detection. Tracking by detection (He et al., 2017) usually employs a detection (or segmentation) step, possibly on all input frames, and later, solves the correspondence problem. These methods can easily take advantage of today's state-of-the-art segmentation (Ronneberger et al., 2015), and can even be trained end-to-end approaches (Wang et al., 2019).

Lens-free digital in-line holographic microscopy (DIHM) has been popular for live cell tracking since the early years of the technology due to its technical simplicity and suitability for resolving small, semi transparent samples (Repetto et al., 2004).

In recent years, deep learning has been used with digital holography in various applications (Rivenson et al., 2019), such as reconstruction: Sinha et al. performed quantitative phase retrieval on a multitude of artificial phase shift images created using a spatial light modulator (Sinha et al., 2017), and the HRNet (Ren et al., 2019) could retrieve phase and amplitude representations of singular objects from actual holograms.

Our approach shows the feasibility of combining tracking by detection with lens-free digital inline holography. A cell-tracking baseline setup uses the angular spectrum method and thresholding to segment white blood cells from human blood. We evaluate whether this step can be replaced by a deep learning approach and show, on simulated and on real holographic images, that CNNs can detect cells accurately and fast by operating on the raw input holograms.

## 2. Methods

Given an in-line holographic microscope, the baseline method for obtaining a statistical analysis of cell behaviour includes five steps:

1.) Capturing holograms (raw data) of a sample, in our case, neutrophils (white blood cells) of humans moving inside a three-dimensional hydrogel, over a specific time-frame in equidistant intervals.

2.) Reconstruction of the hologram for each time step to a phase-shift minimum intensity projection (PSMIP). PSMIPs depict the actual cell shapes in the sample that cannot be seen in the raw data.

3.) Binary segmentation of the PSMIP (3.1), connected component analysis (3.2), and subsequent computation of the components' centroids (3.3).

4.) A combination of single-cell positions over time to cell movement trajectories, also called tracks.

5.) Statistical analysis of those tracks.

In this paper, we train a CNN to eliminate the steps 2 and 3.1. Especially Step 2 is computationally expensive.

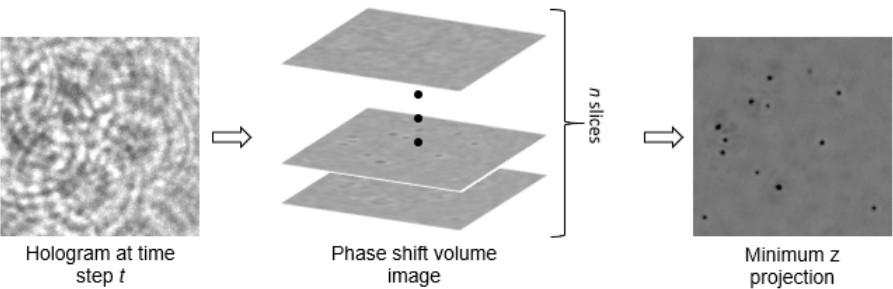

Figure 1: Reconstructing a PSMIP image from a hologram: via the angular spectrum method we focus the raw data (left) to $n$ different positions between sensor and light source (middle). When looking at the phase reconstructions, a dark pixel indicates the position of a cell in the z-axis. For each pixel, we compute the minimum of the $n$ possible values. This leads to the PSMIP image which clearly shows the shape of each individual cell. Note that, for better visibility, only cropped versions of the hologram and reconstruction are shown.

## 2.1. Hologram Acquisition and Reconstruction

For each experiment, several thousand human neutrophils were spread uniformly in a three dimensional hydrogel—mimicking the body's extracellular matrix—where they exhibited spontaneous migration. A custom built lens-less digital holographic in-line microscope was used to record a time-lapse series of 15 captures per minute over the course of 40 to 60 minutes. An example of the raw data with subsequent reconstruction is shown in Figure 1. To the human eye, holograms yield little information since the signals of even small objects appear abstract and spread out in the form of interference patterns. Holographic reconstruction needs to be performed in order to obtain a realistic picture: when specific properties of the device are known, we can calculate an approximation of the complex light field on the optical axis at a desired distance to the sensor plane. We applied the angular spectrum method to create $n = 100 - 120$ reconstructions of each hologram in equidistant intervals of $10\,\mu m$ between two specified distances $z_{start}$ and $z_{end}$ on the z-axis. A real-valued phase shift image was derived from the complex field at each reconstruction distance, effectively creating a volumetric representation of the sample. The lowest intensity values of a given object in the phase shift image can be assumed as the object outline in the focus plane (Sheng et al., 2006). Subsequently, the three dimensional representation of each time step was reduced to a single two dimensional minimum intensity projection:

$$PSMIP(x, y) = \min_{z_1, \ldots, z_n} Vol(x, y, z), \tag{1}$$

i.e., each pixel value represents the lowest intensity along the z-axis $(z_1, \ldots, z_n)$ of the volume $(Vol)$. This step serves to i) move all cells into the same focus plane for easy processing and ii) compress the unwieldy image stacks by a factor of $n$.

## 2.2. Baseline Segmentation, Movement Tracking and Statistical Analysis

Figure 1 shows the reconstruction process, including a PSMIP of a hologram: individual cells can be spotted as small black dots on the image. As the background of a phase shift image remains evenly gray, thresholding is well suited as a quick and robust approach for the separation of object and background pixels. A threshold value of 90 was used to approach the typical neutrophil diameter of around 10 micrometers as given in the literature (Liew and Kubes, 2019). Cell coordinates were then extracted by performing a one-time morphological opening (square kernel with size 2), followed by a connected component analysis and subsequent calculation of the centroids in sub-pixel resolution. From the resulting sets of coordinates for each time step, cell tracks were computed using Trackpy 0.4.2 (Allan et al., 2019), a Python package that implements the popular tracking algorithm proposed by Crocker and Grier. As a general rule, potentially spurious cell tracks with a length shorter than 100 frames (or 400 seconds) were discarded. The average speed of all cells was then calculated for each time step over the course of the experiment. It can be seen as a robust measure for cell behaviour since it does not rely on long-term integrity of the tracks, which can be hampered by collisions.

## 2.3. Dataset

As samples, we collected blood from healthy adult volunteers and isolated the neutrophils. The cells were embedded in a hydrogel and recorded by our holographic device. We collected one image every four seconds for 40 to 60 minutes, creating time-lapse recordings containing 600 to 900 images. We created 14 time-lapse recordings; from those, we used 10 time-lapse recordings as training data for the CNN. From each training time-lapse recording, we selected 20 images at equidistant time points. As our test set, we used the remaining four time-lapse recordings. Note that these recordings differ from the training data since entirely different blood samples were used.

To extend our evaluation to data with perfect labels, we additionally generated simulated holograms based on the baseline segmentations. From each of the four time-lapse experiments, 50 baseline segmentations were chosen in equidistant time intervals. We created a synthetic hologram from each of these 200 binary images by propagating each cell to a random distance that was drawn uniformly from a range that simulated the hydrogel area of the original experiments. The resulting diffraction patterns were added to a single image at their respective real-world xy-positions, creating an artifical hologram of a three-dimensional cell distribution. It should be noted that these simulations can only be seen as an approximation to the experimental data since real-world biological cells diffract incoming light in a more complex way than the binary segmentations representing their outlines.

## 3. Experiments

We ran tests with four different CNN architectures based on the u-net (Ronneberger et al., 2015), but with different Imagenet-pretrained encoders: the ResNet-18, ResNet-50, ResNet-152 (He et al., 2016), and the Mobilenet-V2 (Sandler et al., 2018). The numbers behind the ResNet description indicate the number of layers. The Mobilenet is an architecture specifically designed to be fast and memory efficient. For each model, we used the Pytorch

code from the Segmentation Models Pytorch Github repository (Yakubovskiy, 2020). An example for the raw CNN input is shown in Figure 1 on the left. Note that the hologram was cropped to better show individual cells. The original input patches used during training are larger. Figure 2 shows an overlay that includes a binary CNN output. White and blue pixels show the true positive and false positive predictions of the CNN. Red pixels are false negatives, all other pixels are true negatives.

For simplicity, we employed the same training schedule on all architectures: we trained the network on $512 \times 512$ crops, with a learning rate of 0.01, a batch size of 4, a weight decay of 0.0001 for 50 thousand iteration steps with the Adam optimizer. After 25 thousand iteration steps, we reduced the learning rate to 0.001. The CNN architectures were trained twice: one time for the real holograms and one time for the simulated holograms. For the simulated dataset, we had real ground truth labels. Thus, we also tested the baseline algorithm. Note that none of the real-life training data were used when training the simulated models. They were entirely trained on the simulated data.

We report six different evaluation techniques ((i) - (vi)) to assess different aspects of our proposed approach and the baseline. (i), we evaluated the segmentation quality by computing the Jaccard index (JI) between the binary network prediction ('cell' if Softmax output $\geq .5$) and the label. (ii), we evaluated the cell detection quality of each model: we determined how many of the cells present in the ground truth segmentation were detected in the prediction. Since some of the cells were small, we defined a true positive detection if the bounding boxes of a predicted cell and a ground truth cell had an intersection over union (iou) greater than 0.3. We report the precision, recall, and F1-score of each model. Evaluations (i) and (ii) were done for the real holograms and the simulated dataset. We used a morphological opening with a kernel size of two, to remove noise in the prediction and in the ground truth segmentations. (iii), we estimated the speed of each approach: for each model and the baseline, we estimated the seconds needed for a method to compute the segmentations of 100 holograms, each with a shape of $4225 \times 4225$. These measurements were done on the same machine with an Nvidia RTX 2080ti (11 GB) GPU, an Intel Core i7 8700k (3.7 GHz) CPU and 64 GB RAM.

(iv), one crucial hyperparameter of the detection method is the threshold that is applied to the probability output of the network. Thus, we expanded our experiments to quantify the variation caused by different threshold values: we redo the measurements (i) and (ii) for the thresholds 5 %, 25 %, and 95 % with the ResNet-18 on the real holograms.

To evaluate if the CNN-based tracking results differ from the baseline tracking approach, we used the proposed tracking algorithm (see Section 2.2) with both the CNN segmentation and the baseline. (v), from the resulting tracks, we computed the average speed curves for each of the four test stacks. The respective baseline and CNN curves were compared using the Pearson correlation coefficient (PCC).

(vi), another way to evaluate the soundness of the approach was to determine whether the results are realistic. From our data, we knew that the number of cells should be constant over an entire time-lapse recording. To test if this behavior was present in our approaches, we computed a linear fit of the curve (x: time, f(x): number of cells). We did this for the ResNet-18 with different thresholds and the baseline. Arguably, for a practical method, the linear function's slope should be close to zero.

Table 1: Results for the simulated data.

| CNN | Prec. | Rec. | F1 | JI |
|---|---|---|---|---|
| Mobilenet-V2 | 0.981 | 0.952 | 0.966 | 0.844 |
| ResNet-18 | 0.976 | 0.959 | 0.967 | 0.845 |
| ResNet-50 | 0.979 | 0.956 | 0.967 | 0.832 |
| ResNet-152 | 0.975 | 0.961 | 0.968 | 0.847 |
| Baseline | 0.884 | 0.860 | 0.870 | 0.664 |

Table 2: Results for the segmentation evaluation of different CNN architectures: the numbers show the mean values of a stack (400-800 time-lapse images) averaged over all 4 test stacks. Additionally, the inference time for 100 images with a resolution of $4225 \times 4225$ is given. The baseline approach needed 392 seconds.

| CNN | Prec. | Rec. | F1 | JI | sec |
|---|---|---|---|---|---|
| Mobilenet-V2 | 0.897 | 0.772 | 0.829 | 0.605 | 115 |
| ResNet-18 | 0.941 | 0.869 | 0.903 | 0.696 | 119 |
| ResNet-50 | 0.929 | 0.852 | 0.889 | 0.673 | 172 |
| ResNet-152 | 0.947 | 0.849 | 0.895 | 0.669 | 246 |

## 4. Results

Table 1 shows the segmentation results for the four network architectures and the baseline on the simulated dataset. With a JI above 80 %, all CNNs were capable of predicting accurate segmentations. The baseline was less accurate with a JI of 66 %. Note that we tested different global thresholds to compute the baseline segmentation and we report the best results we found. The CNNs detected almost all cells (recall over 95 %) with only a few false positives (precision over 97 %). Table 2 shows the segmentation results for the four network architectures that we evaluated on the real holograms. With an F1-score of over 80 %, all four models could perform accurate cell detection. Interestingly, the ResNet-18 reached a 90 % score, which is a remarkable result since deeper models usually perform with higher accuracy. However, we did not use an extensive hyperparameter search, and

Table 3: Results for different binarization thresholds for the ResNet-18: the slope of the ground truth value was $-0.250$.

| Thres. | Prec. | Rec. | F1 | JI | Slope | PCC |
|---|---|---|---|---|---|---|
| .05 | 0.845 | 0.900 | 0.871 | 0.622 | $-0.039$ | 0.989 |
| .25 | 0.921 | 0.918 | 0.919 | 0.724 | $-0.211$ | 0.989 |
| .50 | 0.941 | 0.869 | 0.903 | 0.696 | $-0.435$ | 0.988 |
| .95 | 0.684 | 0.403 | 0.505 | 0.324 | $-1.231$ | 0.974 |

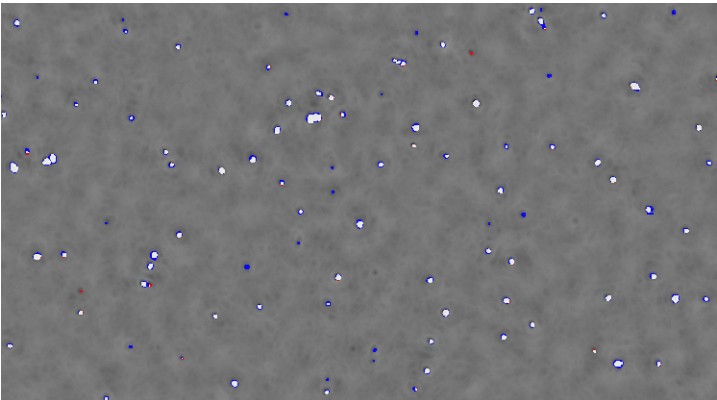

Figure 2: Comparison of the network output to the baseline algorithm: the image shows the PSMIP of a test image with white pixels being true positives, red pixels being false negatives, and blue pixels being false positives. However, images like this indicate that many of the false-positive detections seem to be actual cells. Furthermore, statistical analysis shows that many of those cells show similar behavior to ground truth cells.

configurations may exist that yield more accurate results for the deeper models. Compared to the simulations, the JI was decreased from over 80 % to 60-70 %. This gap exists because the labels of the real dataset did not reach the quality of the simulated data. To acquire the labels for the real holograms, we used the baseline approach. However, this baseline may not have correctly detected all cells, which lead to noisy labels. Furthermore, the simulated holograms contained only cells. On the other hand, the real holograms did also show the outlines of the hydrogel or artifacts coming from dirt. Nevertheless, our results showed that segmentation of cells in holograms was feasible with the rather small ResNet-18, which was also the 2nd fastest of the architectures, only outperformed by the Mobilenet. In direct comparison to the baseline algorithm, we saved 273 seconds for the segmentation of 100 images; thus, our approach was three times faster. The results in Table 3 show that the detection recall and precision could be controlled with the binarization threshold. With a threshold of .25, the F1-score was increased from 90.3 % to 91.9 %. However, the precision and recall values indicated that, even for our best model, the number of detected cells could differ considerably from the baseline. If these false positive and false negative cells were really mistakes made by the CNN, for example, artifacts that were wrongly segmented or cells that were overlooked, one would expect to see a different CNN tracking curve compared to the baseline. However, when looking at Table 3, we see a strong PCC between the predictions of the ResNet-18 and the reference average speed curves. This quantifies that the overall behavior we observed with our baseline method could be reproduced with our deep-learning approach. Furthermore, upon visual inspection, the 'false positives' detected by the deep network appeared to be valid cells that were missed by the baseline approach (example shown in Figure 2). In addition, Table 3 shows that using the ResNet-18 with a threshold of .05 lead to a constant cell number over time. This output was the most realistic since we knew from our data that no considerable change in the cell number occurred.

Although the F1-score decreased to 87 %, there was still a high correlation to the baseline average speed curve. For these three reasons (no change in cell behavior, visual inspection, and constant cell number), it is likely that the ResNet-18 with a threshold of .05 performed better than the ground truth.

## 5. Conclusion

We have presented a simple, fast, and accurate framework for cell-tracking with lens-less devices. Our results show that even shallow networks, such as the ResNet-18, perform well when applied directly to the raw holographic data, thus avoiding the more costly process of first reconstructing images from the raw data. On simulated data, the CNNs yield a near perfect result. When segmentation on real data is performed with high confidence, e.g., a 25 % threshold, the segmentation results of the baseline model (that employs image reconstruction) can be reproduced. With lower thresholds, the reproduction is less accurate and the direct method detects more cells than the baseline method. Visual inspection and statistical analysis, however, reveal that these additional cells are valid and move with the expected velocities. Moreover, as one would expect, the number of tracked cells does not decrease. Thus, one can conclude that direct inference based on deep learning outperforms the more complex model of first reconstructing images based on the angular spectrum method (it is easier to segment a cell without seeing one).

## 6. Compliance with Ethical Standards

Blood collection was conducted with the agreement and written consent form of each participant and was approved by the ethical committee of the University of Lübeck (18–186).

## Acknowledgments

The authors thank Mr. Prof. Dr. Tamás Laskay, Ms. Mareike Ohms and Mr. Daniel Dömer for constructive discussions and provision of primary human neutrophils. This work was partly funded by the European Regional Development Fund (ERDF).

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
