# OpenReview forum: "Direct Inference of Cell Positions using Lens-Free Microscopy and Deep Learning"
_MIDL.io/2021/Conference — MIDL 2021_

### Official Review · AnonReviewer1 · 2021-03-05

**Confidence:** 3
**Preliminary Rating:** 2
**Recommendation:** Poster
**Final Rating:** 3

**Summary:**

The authors present a cell position detection algorithm that directly works on holographic data. As the model works directly on holographic data, it eliminates the need for computationally expensive image reconstruction steps. Authors show experimental results on simulated data as well as real data (from blood collected from healthy adults).

**Strengths:**

The application makes a lot of sense. It is similar to the old ideas where people were trying to detect detection in compressed videos without decompressing videos. If the ultimate goal is just extracting the cells' raw locations in an image, then this application is highly beneficial. This can find good use in lab settings where actual images are not needed.

**Weaknesses:**

Doing all the processing in the hologram data domain makes the model not interpretable as the user can not see the images. The technical novelty is limited as the authors used readily used tools from the literature to achieve their goal. Nothing specific to the given application is developed. Conclusions are weakly based on the results.

**Deanonymize Review:**

no

**Final Rating Justification:**

Thanks to the authors for their detailed response and for clearing some of the confusion in the first version of the article. I've updated my score in light of the new details that the authors provided.

**Justification Of The Preliminary Rating:**

Even if the application is fairly novel, in terms of technical novelty, the paper is limited
The conclusions are weakly supported by results.  It is not clear how the authors reached the presented conclusions
The gap between the simulated and the real data results is very large. The reason for this gap is not explained.


**Paper Type:**

validation/application paper

**Questions To Address In The Rebuttal:**

1) Image reconstruction from holographic data is a well-studied topic in the literature. The authors also mentioned the method of Rivenson, who also uses a DNN for rapid reconstruct of images from holographic data. In the reviewer's opinion, authors should have compared their method against image reconstruction using Rivenson methods followed by binary segmentation (which has very low computational complexity anyways)
2) It is not clear how the training was conducted on the 200 simulated images mentioned in the paragraph before the experiments section. Did the authors train a brand new model for this data, or they added this data to their real-life data  (described in the first paragraph of the datasets section)  and retrained their original model?
3) Gap between the results given in Table1 (real-life data) and Table 2 (Simulated data) is very large. What is the reason, and how can the gap be closed
4) In the results section, the authors state that " the network likely performed better than the ground truth." This claim must be supported concretely with data. Do they have any quantitative data to support this claim?
5) Again the same section, the authors state that the network performed very well if Table 2 results are taken as a base. On the other hand, the gap between tables 1-2 can also be a sign of lack of generalizability and/or model overfit to the clean data.
6) In the conclusions section, the authors state that; "Visual Inspection and statistical analysis, however, reveal that these additional cells are valid and more with the expected velocities" However, only visual results were presented. No statistical analysis was presented in the results section.
7) It is not clear how the authors reached the conclusions stated at the end of page 7.
8) Overall, the conclusions are weakly supported by the results.

**Special Issue:**

no

---

> ### Author Response · Authors · 2021-03-17
> **Response to AnonReviewer1**
>
> *Image reconstruction from holographic data is a well-studied topic in the literature. The authors also mentioned the method of Rivenson, who also uses a DNN for rapid reconstruct of images from holographic data. In the reviewer's opinion, authors should have compared their method against image reconstruction using Rivenson methods followed by binary segmentation (which has very low computational complexity anyways)*
>
> In Rivenson’s review, only Sinha’s method is likely to match the speed of our method. However, the quality of the reconstructed images is not suitable for the segmentation of cells.
>
>  *It is not clear how the training was conducted on the 200 simulated images mentioned in the paragraph before the experiments section. Did the authors train a brand new model for this data, or they added this data to their real-life data (described in the first paragraph of the datasets section) and retrained their original model?*
>
> From the original baseline segmentations, we created simulated holograms. We created an entirely new training data set and a new test data set. Furthermore, for each of the four tested architectures, we trained a new model. Thus, the ResNet-18 on the simulated data did not see any original data, and vice-versa. We added an explanation to the text.
>
>
>
> *Gap between the results given in Table1 (real-life data) and Table 2 (Simulated data) is very large. What is the reason, and how can the gap be closed?*
>
> The labels used to train the networks in the real-life data are far noisier: our results indicate that the baseline algorithm, i.e., the reconstruction with thresholding, creates false negatives. However, we use the baseline outputs as labels, thus, creating an imperfect dataset to train the CNNs. For the simulated data, there are no noisy labels, as well as no artifacts.  One could close the gap by using hand-labeled real-life data, but they are hard to acquire. A small number of labeled data in addition to a large number of simulated data may work. In the revised manuscript, we commented on the gap between the results. We thank the reviewer for pointing out this unclear paragraph.
>
>
> *In the results section, the authors state that " the network likely performed better than the ground truth." This claim must be supported concretely with data. Do they have any quantitative data to support this claim? Again the same section, the authors state that the network performed very well if Table 2 results are taken as a base. On the other hand, the gap between tables 1-2 can also be a sign of lack of generalizability and/or model overfit to the clean data. In the conclusions section, the authors state that; "Visual Inspection and statistical analysis, however, reveal that these additional cells are valid and more with the expected velocities" However, only visual results were presented. No statistical analysis was presented in the results section.*
>
> The CNNs detect cells that are not labeled as cells in the ground truth. These cells cannot be false positives for the following three reasons: i.) they look like cells, ii.) the average speed curves computed by the baseline are similar to the average speed curves computed with the CNNs, and iii.) with the right threshold, the number of cells remains constant. Point ii.) was quantified using the Pearson correlation coefficient between the baseline average speed curves and the curves computed by the CNN. Point iii.) was quantified using the slope of the number of cells over time. Using the ResNet-18 and a threshold of 0.05, the number of cells was almost constant, while the average speed curves had a high correlation with the baseline curves.
> We thank the reviewer for the valuable insights.

---

### Official Review · AnonReviewer3 · 2021-03-08

**Confidence:** 5
**Preliminary Rating:** 2
**Recommendation:** Poster

**Summary:**

The paper describes a cell tracking technique for time lapse lens-free microscopy images. The images are reconstructed from hologram using phase-shift minimum intensity projection. They used U-net architecture to generate segmentation maps, then, cells are tracked using Trackpy package. They studied different CNN backbones for that task.

**Strengths:**

They compared different backbones for the U-net architecture and they also eliminated two steps from the conventional methods for the cell tracking in time lapse holograms. They backed their results using quantitative results for tracking and segmentation.

**Weaknesses:**

The authors replaced the u-net architecture to replace reconstruction using  phase-shift minimum intensity projection. However, they did not compare their approach to a convention method that do reconstruction using CNN then do the binary segmentation for the reconstructed technique.

They claimed that in the conclusion that the proposed is fast. Is it fast in terms of less steps or number of layers compared to other reconstruction methods?

**Deanonymize Review:**

no

**Detailed Comments:**

- A block diagram to show the inputs and the output to the network is crucial.
- The synthesis process of the test data set is not clear.
- In Table 3, they used different binarization threshold for the ResNet-8. They did not do the same study on the baseline too. They should do the same. Also, compare it to the results of the reconstruction using CNN to show the effect.

**Justification Of The Preliminary Rating:**

The author proposed a cell tracking technique using CNN and they applied different backbones for U-net architecture. However, they failed to compare the proposed work to the existing method other than the baseline and they claimed that it is faster however, they did not compare execution times.

**Paper Type:**

validation/application paper

**Questions To Address In The Rebuttal:**

- What is the advantage of this method over reconstruction using the CNN then binary segmentation?

**Special Issue:**

no

---

> ### Author Response · Authors · 2021-03-17
> **Response to AnonReviewer3**
>
> *The authors replaced the u-net architecture to replace reconstruction using phase-shift minimum intensity projection. However, they did not compare their approach to a convention method that do reconstruction using CNN then do the binary segmentation for the reconstructed technique.*
>
> We agree with the author that a machine learning approach on the actual reconstructions may yield better results. However, the training labels are generated from the thresholding of reconstructed images. We do not have hand-labeled data at our disposal. If we use the current labels, all the CNN would need to learn on the reconstruction is to imitate the threshold function. Furthermore, combining the reconstruction (a computationally expensive operation) with a CNN (another computationally expensive operation) is not a comparative baseline, since it is even slower than the thresholding approach.
>
> *They claimed that in the conclusion that the proposed is fast. Is it fast in terms of less steps or number of layers compared to other reconstruction methods?*
>
> We reported the number of seconds needed to segment 100 holograms on the same machine. From the original manuscript:
> “ For each model and the baseline, we estimated the seconds needed for a method to compute the segmentations of 100 holograms, each with a shape of 4225×4225. These measurements were done on the same machine with an Nvidia RTX 2080ti (11 GB) GPU, an Intel Core i7 8700k (3.7 GHz) CPU, and 64 GB RAM. [...] Nevertheless, our results showed that segmentation of cells in holograms was feasible with the rather small ResNet-18, which was also the 2nd fastest of the architectures, only outperformed by the Mobilenet. In direct comparison to the baseline algorithm, we saved 273 seconds for the segmentation of 100 images; thus, only a third of the time was needed with our new approach.”
>
> *A block diagram to show the inputs and the output to the network is crucial.*
>
> The network input is depicted in Figure 1 on the left. The output of the network is shown as an overlay with the phase-minimum projection in Figure 2. Since this was not obvious in the original manuscript, we added an explanation in the text.
>
> *The synthesis process of the test data set is not clear.*
>
> We updated the text, explaining the simulation process, including how the simulated dataset was generated.
>
> *In Table 3, they used different binarization threshold for the ResNet-8. They did not do the same study on the baseline too. They should do the same. Also, compare it to the results of the reconstruction using CNN to show the effect.*
>
> We agree with the author. We used different global thresholds for the baseline segmentation on the simulated dataset and reported the best result. We mention this in the new manuscript.
>
> *What is the advantage of this method over reconstruction using the CNN then binary segmentation?*
>
> The significant increase in speed is a massive benefit for the application. Furthermore, the reconstruction may introduce additional errors.
>
> *The author proposed a cell tracking technique using CNN and they applied different backbones for U-net architecture. However, they failed to compare the proposed work to the existing method other than the baseline and they claimed that it is faster however, they did not compare execution times.*
>
> We indeed only compare with the baseline, including execution times with results shown in Table 1. We thank the reviewer for the comments.

---

### Official Review · AnonReviewer4 · 2021-03-08

**Confidence:** 4
**Preliminary Rating:** 3
**Recommendation:** Oral, Poster
**Final Rating:** 3

**Summary:**

The authors present a framework for tracking cells in lens-free digital in-line holography. In their method, they describe how to bypass the computationally intensive image reconstruction step with a deep neural network that can directly predict cell masks. The authors then study the increase in performance and the quality of the results.

**Strengths:**

- The presented work addresses an interesting approach to improve label-free cell tracking (at least in terms of speed).
- The work is well motivated and the paper is (in most parts) well written and easy to follow.


**Weaknesses:**

- The major drawback is the missing manual ground-truth here. While comparing the segmentation and tracking results with a baseline computational result is surely a reasonable first approximation (also here, a more sophisticated approach such as CellPose [1] might be a better choice). However, a real manually validated reference would be tremendously helpful to understand the benefits or drawbacks of the approach.
    - 1. Stringer, C., Wang, T., Michaelos, M. & Pachitariu, M. Cellpose: a generalist algorithm for cellular segmentation. __Nature Methods__ **18**, 100–106 (2020).
- The simulation of training data is not well explained. More details are needed, since this is the only hard ground-truth that is available for assessing the results.
- The improvements in speed seem rather moderate. How much is needed for practical purposes? This is hard to put in context with the information given so far.
- The assessment of tracking accuracy is based on linear fits to cell numbers over time. This is a quite an indirect measure that somewhat complicates the discussion. I would suggest reporting object-level results instead (false splits and merges) for each frame in addition.
- The results part feels a bit unstructured. This should be revised and written more clearly. Maybe having a better quantitative measure would help to sharpen the discussion.


**Deanonymize Review:**

no

**Final Rating Justification:**

Thanks a lot for answering the questions and clarifying the presentation in the paper.

**Justification Of The Preliminary Rating:**

I like this paper in principle. Label-free tracking is an interesting and relevant biological method and this paper presents a reasonable first step to improving existing methods. But as it stands, the paper feels a bit unfinished. The results part needs more thought and revision.

**Paper Type:**

both

**Questions To Address In The Rebuttal:**

- explain simulation in more-depth
- revise results and validation part (maybe focus more on simulated data)


**Special Issue:**

no

---

> ### Author Response · Authors · 2021-03-17
> **Response the AnonReviewer4**
>
> *The major drawback is the missing manual ground-truth here. While comparing the segmentation and tracking results with a baseline computational result is surely a reasonable first approximation (also here, a more sophisticated approach such as CellPose [1] might be a better choice). However, a real manually validated reference would be tremendously helpful to understand the benefits or drawbacks of the approach.*
>
> A manual ground truth would be very helpful but challenging to create since the holograms cannot be segmented manually, and the reconstruction would introduce errors.
>
>
> *The simulation of training data is not well explained. More details are needed, since this is the only hard ground-truth that is available for assessing the results.*
>
> We agree with the author. We revised the Method Section accordingly.
>
> *The improvements in speed seem rather moderate. How much is needed for practical purposes? This is hard to put in context with the information given so far.*
>
> The reconstruction is by far the most time-consuming step of the pipeline, with execution time in the realm of hours for a real-world experiment (500-900 holograms). Therefore, for practical purposes, even a moderate speedup would be helpful. However, we are three times faster, which is a significant speed-up.
>
> *The assessment of tracking accuracy is based on linear fits to cell numbers over time. This is a quite an indirect measure that somewhat complicates the discussion. I would suggest reporting object-level results instead (false splits and merges) for each frame in addition.*
>
> We only evaluate the cell number over time to argue that our model is more consistent than the ground truth because we know, based on the design of the experiment, that the number of cells should be constant. We have the text to avoid this misunderstanding.
>
> *The results part feels a bit unstructured. This should be revised and written more clearly. Maybe having a better quantitative measure would help to sharpen the discussion.*
>
> We revised the results section. We thank the reviewer for the constructive and precise comments.

---

### Official Review · AnonReviewer2 · 2021-03-08

**Confidence:** 4
**Preliminary Rating:** 2
**Recommendation:** Poster
**Final Rating:** 2

**Summary:**

The submitted work covers the problem of automatically localizing cells in lens-free digital in-line holography images using a trained CNN. Both semi-automatically generated annotations and computational simulations are employed to study a U-Net with different standard pre-trained encoders (MobileNet and ResNets). The authors compare their results using common classification accuracy measures and evaluating the result in the tracking of the detected cells.

**Strengths:**

- The authors show a new image processing through DL aplication set-up.

- In the framework of cell tracking, the authors show that the cell detection step in this setup can be successfully conducted using known pre-trained architectures rather than performing image reconstruction, which is more laborious and not completely solved.

- They use semi-automatically annotated images and simulations as ground truth. Visual inspection of the results shows that the network could be able to generalize and be more accurate than the annotations performed to train it.



**Weaknesses:**

- When reading the abstract, it seems that the authors will show a method for cell tracking. However, the work is specially focused on accurately detecting cells.

- "After one third and two-thirds of the training time, we reduced the learning rate to 0.01 and 0.001, respectively." Please, be more specific.

- Is the test set the same when training the networks with annotated data and the simulated data?

- It seems that most trained networks would perform the task accurately, however, a fair comparison would imply some kind of cross-validation for each architecture and training schema.

**Deanonymize Review:**

no

**Final Rating Justification:**

I thank the authors for the rebuttal. All the questions were properly answered and clarified. However, I still think that there is little novelty in the presented work.

**Justification Of The Preliminary Rating:**

While the work is well written and the authors present a new application of DL, I think the work lacks novelty with respect to image processing techniques or DL training theory in biomedical image analysis.

**Paper Type:**

validation/application paper

**Special Issue:**

no

---

> ### Author Response · Authors · 2021-03-17
> **Response to AnonReviewer2**
>
> *When reading the abstract, it seems that the authors will show a method for cell tracking. However, the work is specially focused on accurately detecting cells.*
>
> We revised the abstract to avoid this misunderstanding.
>
> *"After one third and two-thirds of the training time, we reduced the learning rate to 0.01 and 0.001, respectively." Please, be more specific.*
>
> We revised this Section.
>
> *Is the test set the same when training the networks with annotated data and the simulated data?*
>
> The simulated data stem from the original data’s baseline segmentations. I.e., to create a simulated image, we transform the segmentation back to a raw image. To make the entire simulated test set, we only use segmentations from the original test set. However, the raw data which are passed to the network during testing are still simulated.
>
> *It seems that most trained networks would perform the task accurately, however, a fair comparison would imply some kind of cross-validation for each architecture and training schema.*
>
> We agree that some of the deeper CNNs may benefit from an included hyperparameter search. Thus, they may perform better than the ResNet-18 if specific parameters, such as another learning rate, weight decay, dropout, or a different optimizer, are used. However, one of the critical factors of the approach is improved speed. Hence, if we reach slightly better results with a deeper network, the ResNet-18 would still be the preferred architecture as it is accurate and fast. Concerning the use of cross-validation, the number of images, and more importantly, the number of cells, is already very large in the used dataset. Thus, it is unlikely that cross-validation would lead to different results.
>
> *While the work is well written and the authors present a new application of DL, I think the work lacks novelty with respect to image processing techniques or DL training theory in biomedical image analysis.*
>
> We thank the reviewer for the thorough and insightful comments. Indeed, the novelty lies in the application and in the fact that we can directly segment cells in the holograms. We think that this paper is a good fit for this conference since the MIDL explicitly offers the paper type ‘application’.

---

### Meta-Review · Area_Chairs · 2021-03-31

**Recommendation:** Accept (Poster)

**Metareview:**

the reviewer's opinion was split on this paper, but after rebuttal no major concerns remained so I recommend acceptance. However, the authors should aim to incorporate the answers given to the reviewers concerns in the final version.

**Paper Type:**

validation/application paper

---

### Decision · Program_Chairs · 2021-03-31

Accept